# Exploring Microeukaryote Community Characteristics and Niche Differentiation in Arid Farmland Soil at the Northeastern Edge of the Tibetan Plateau

**DOI:** 10.3390/microorganisms11102510

**Published:** 2023-10-08

**Authors:** Lingyun Chen, Haifeng Han, Chunhui Wang, Alan Warren, Yingzhi Ning

**Affiliations:** 1College of Life Science, Northwest Normal University, Lanzhou 730070, China; lychen@nwnu.edu.cn (L.C.); 2020212611@nwnu.edu.cn (H.H.); 2020212604@nwnu.edu.cn (C.W.); 2Department of Life Sciences, Natural History Museum, London SW7 5BD, UK; a.warren@nhm.ac.uk

**Keywords:** agricultural soil, beta analysis, co-occurrence network

## Abstract

The northeastern edge of the Tibetan Plateau exhibits diverse climate and landform variations, and has experienced substantial recent environmental changes, which may significantly impact local agricultural practices. Understanding the microeukaryote community structure within agricultural soils is crucial for finding out the biological responses to such changes and may guide future agricultural practices. In this study, we employed high-throughput amplicon sequencing to examine 29 agricultural soil samples from seven research areas around the northeastern edge of the Tibetan Plateau. The findings revealed that the predominant biological communities in these soils were characterized by a high abundance of Alveolata, Amoebozoa, and Rhizaria. Ascomycota displayed the highest relative abundance among fungal communities. Moreover, notable distinctions in microeukaryote community composition were observed among the study sites. Co-occurrence network analysis highlighted interactions between the biological communities. Furthermore, our results elucidated that deterministic and stochastic processes exerted diverse influences on the distribution of protozoan and fungal communities. This study provides valuable insight into the microeukaryote structure in the agricultural soils of the northeastern edge of the Tibetan Plateau, shedding light on the intricate relationships between environmental factors, microeukaryote communities, and agricultural productivity.

## 1. Introduction

Soil organisms, i.e., any organism inhabiting the soil during part or all of its life and which range in size from microscopic cells that digest decaying organic material to small mammals that live primarily on other soil organisms, play an important role in maintaining the fertility, structure, drainage, and aeration of soil [1]. The soil ecosystem harbors an intricate and diverse array of communities, comprising millions of species and billions of individual organisms, which collectively shape its functionality [2]. A comprehensive understanding of the soil food web necessitates the inclusion of data on all groups of soil organisms, as their presence is vital for a holistic comprehension of ecosystem dynamics. Soil organism communities exhibit remarkable diversity and abundance, and play essential ecological roles [3]. Soil, as the foundation of agricultural development, relies on the health and vitality of its resident organisms [4]. Therefore, the promotion of sustainable agriculture hinges upon recognizing the positive influence exerted by soil organisms as crucial drivers of soil quality [5,6]. Moreover, soil biodiversity assumes a pivotal role in suppressing soil-borne diseases, thus facilitating food and water security and environmental quality [7].

The concept of soil health entails the continuous capacity of soil to function as a vibrant living ecosystem that sustains plants, animals, and humans [8]. Every chemical transformation occurring within soil involves active contributions from its diverse array of organisms [9,10]. Unfortunately, unsustainable agricultural practices and intensive farming have led to widespread soil degradation, loss of soil biodiversity, carbon and nutrient depletion, food insecurity, and risks to human health [11,12,13,14]. Soil organisms significantly influence plant growth and soil fertility [15,16]. Therefore, assessing the current status of soil organisms assumes great significance for safeguarding soil biodiversity and promoting sustainable agricultural development [17,18].

The productivity and well-being of agricultural systems are intrinsically linked to the functional processes facilitated by soil microbial communities [13,14,16]. Unraveling the complexities of soil microbial communities in arid farmlands will ultimately contribute to informed decision making regarding sustainable agricultural practices and soil ecosystem preservation. Accordingly, this study explores the microbial communities in arid farmlands within China. The research area is situated at the confluence of the Loess Plateau and the Qinghai−Tibet Plateau, which is characterized by minimal precipitation and severe soil erosion. The primary objective of this study is to investigate the diversity and community structure of soil microeukaryotes, including protists, and fungi, while considering their interactions in relation to different crops and geographical locations. Employing soil collection in seven sampling areas and using high-throughput amplicon sequencing of partial amplification of the 18S rRNA gene, we aim to assess and compare biodiversity and community composition variations arising from crop growth and climatic conditions. Additionally, co-occurrence network analysis will shed light on the intricate interactions among these organisms.

## 2. Materials and Methods

### 2.1. Sampling Areas and Site Descriptions

Gansu Province is in the northwest of China at the intersection of three major plateaus (Loess Plateau, Qinghai Tibet Plateau, and Inner Mongolia Plateau) and three major natural areas (the northwest arid region, the Qinghai−Tibet Plateau, and the eastern monsoon region). Its geological landforms and weather patterns are complex and diverse, and various geological forms, including plateau, mountain, desert, and the Gobi Desert, are intertwined. Previous studies have focused on the central region of the Tibetan Plateau, but our study area is located in its northeastern margin, which complements systematic studies of this plateau. A comprehensive dataset comprising 87 soil samples was collected (Table 1), representing 29 different crop soils. These samples were geographically divided into seven sampling areas, as depicted in Figure 1.

Figure 1 was constructed using ArcMap 10.8 with the base bitmap downloaded from the National Catalogue Service of Geographic Information [19] and Geospatial Data Cloud [20].

### 2.2. Soil Sample Collection and Storage

Soil samples were collected in different research areas in August 2018 (Figure 1). Three 5 m × 5 m parallel quadrats were set up for each crop type, and the samples were taken in three independent replicates from each sampling site. The surface soil (0 cm to 5 cm) was collected strictly according to the five-point sampling method, and the fresh soil samples were stored in plastic bags. Samples were stored at minus 20 °C to prevent DNA degradation. The soil temperature and geographic location were recorded at each collection point (Table 1). 

### 2.3. DNA Extraction and PCR Amplifications

The extraction of environmental DNA from soil was accomplished through the direct lysis method according to Novogene Science and Technology Company (Beijing, China). To obtain sufficient DNA, extraction was carried out in 30 replicates, and the DNA was pooled and concentrated. The composition and diversity of protists and fungal communities in the soil were determined by amplifying the V4 regions of 18S ribosomal RNA (rRNA) genes. The primers 528 F (5′-GCGGTAATTCCAGCTCCAA-3′) and 706 R (5′-AATCCRAGAATTTCACCTCT-3′) and the DNA extracted from the samples as templates were used for amplification [21].

### 2.4. High-Throughput Sequencing

In short, PCR products were purified and visualized on a 2% agarose gel. PCR products were mixed in equidensity ratios. The mixture of PCR products was purified using the GeneJETTM Gel Extraction Kit (Thermo Scientific, Waltham, MA, USA). The library quality was assessed using a Qubit@ 2.0 Fluorometer (Thermo Scientific, Waltham, MA, USA). The prepared DNA libraries were pooled and sequenced on an Ion S5TM XL platform using 600 bp single-end reads V4 chemistry. All of these steps were completed by the Novogene Science and Technology Company (Beijing, China).

### 2.5. Data Processing

Raw reads were assigned to original samples according to the barcode and were truncated by cutting off the barcode and primer sequences using FLASH (V1.2.7) [22]. Raw data with low-quality nucleotides (*Q*-value < 19) were eliminated using QIIME [23]. Operational taxonomic units (OTUs) clustering was performed at a cut-off value of 97% sequence similarity using QIIME (Version 1.9.1) [23]. Singletons were removed and chimeric sequences were screened out through Vsearch [24]. One representative sequence from each OTU was BLAST searched against the Silva 18S rRNA database (SSU 132) with a confidence threshold of 60% for taxonomic affiliations [25,26]. OTUs affiliated with metazoan and terrestrial plants were removed from the dataset.

### 2.6. Statistical Analysis

The dataset was subsampled to normalize the number of reads per sample using the criterion of the least amount of data in the sample. The subsequent alpha diversity analysis and beta diversity Analysis were based on the homogenized data. 

The geom_bar function in the ggplot2 package was used to draw the histogram of the relative abundance of the soil microeukaryotes [27]. The “position” parameter was selected as “fill”, which meant stacking the elements and standardizing them as “1”. In our study, we followed Tikhonenkov et al., and concluded that the alveolates included the dinoflagellates, about half of which were algae with complex plastids, and two large and important groups of protozoa: the Apicomplexa, and the mostly free-living ciliates, and the Colponemidia, of the sister lineage to all other known alveolates [28]. The analysis of the microeukaryotic composition was accomplished based on OTUs at the phylum level.

Two diversity parameters, i.e., species diversity (Shannon-Weaver H′) and dominance index (Simpson C′), of the protozoa and fungal communities were calculated using R software with the “vegan” package, and the significance was tested using Wilcoxon test [29]. R v.4.2.2 (package ggpolt2) was used for nonparametric multidimensional analysis (based on the Bray−Curtis distance). The alpha and beta diversity of the protists and fungi in the samples were assessed using genus-level OTUs.

Co-occurrence patterns were revealed through network inference using strong and significant correlations based on non-parametric Spearman’s analysis [30]. Spearman’s rank correlations (r) among the eukaryotes were calculated within the “psych” R package. Statistically significant (*p*-value ≤ 0.05) correlations were incorporated into the network analyses [31]. Network visualization was drawn using Gephi version 0.9.6 [32,33]. The symbiotic networks were constructed using the same thresholds (*r* > 0.6, *p* ≤ 0.05) to facilitate comparative analysis.

Niche breadth was calculated with RStudio using the “niche. width” function within the “Spaa” R package [34,35]. A neutral community model (NCM) was constructed using “Hmisc”, “minpack. lm”, and “stats4” R packages [35,36,37]. The data analysis and data visualization described above were based on OTU data at the genus level.

## 3. Results

### 3.1. Taxonomic Compositions of the Soil Microeukaryote Communities in Different Sampling Areas

The relative abundance of soil protist and fungal communities showed obvious variations among the seven sampled regions (Figure 2A,B respectively). The relative abundance of protists in the seven regions is shown in Figure 2A using a stacking column diagram. Alveolates were found in a significant proportion in all regions, accounting for 50% in each region, with the highest abundance being observed in WD and the lowest in LX. Amoebozoa and Rhizaria were also present in all of the regions, with their proportions being notably higher than the other protist groups. Additionally, Archaeplastida, Opisthokonta, and Excavata were each found in only a few specific areas.

There were evident differences in the fungal communities in the seven regions (Figure 2B). Asomycota was the dominant group in all seven regions, accounting for more than 50% of the entire fungal community, with the highest relative abundance of species in the HN region, accounting for more than 90%. Mucoromycota was also detected in all regions, albeit with an obviously lower proportion than Asomycota, except in DYG where the two groups exhibited nearly identical relative abundances. Aphelida species were only found in WD and TC but with low relative abundance. The relative abundances of Aphelida, Cryptomycota, unidentified fungi and Zoopagomycota were very low in all sampled regions. 

### 3.2. Alpha Diversity of the Soil Microeukaryote Communities

Figure 3 illustrates the variations in soil biodiversity indices (Shannon and Simpson) across different sampling areas. Notably, significant and discernible differences in soil protist and fungal communities were observed between certain areas at the genus level. The Shannon diversity of soil primary communities in DYG, TC, WD, and LX was significantly higher compared with DX (Figure 3A, *p* ≤ 0.05), especially in DYG (*p* ≤ 0.001) and LXL (*p* ≤ 0.01). There was no significant difference in the diversity of communities between HN and TC. The difference in the dominance index among the study areas was almost similar to the diversity index, except between LX and LXL. There was no statistically significant distinction in the dominance of communities between LX and LXL (*p* > 0.05).

The diversity of the fungal communities was not completely consistent with that of the protist communities (Figure 3B). The analysis shows that the fungal community diversity did not exhibit a statistically significant difference between DX and WD. However, it is worth noting that the community dominance of DX was significantly higher than that of WD (*p* ≤ 0.05). There was no significant difference in fungal community diversity and community dominance between DX and HN (*p* > 0.05). The study reveals a notable distinction in both diversity and dominance within the fungal community of LX in comparison with DYG and WD. 

### 3.3. Beta Diversity of the Soil Microeukaryote Communities

The difference in geographical environment had a very significant effect on the community structure of the soil protists and soil fungi. As shown in Figure 4, it is clear that the structure of the soil protist community differed significantly in different regions, and the Adonis (R^2^ = 0.5, *p* ≤ 0.001) and Anosim (R = 0.31, *p* ≤ 0.01) test structures also indicated this. There was, however, disparity in the significance of the distinctions depicted by the two test methods. Nevertheless, it is important to highlight that DYG and LX exhibited similarities in the structure of their communities. On the other hand, the analysis of variance shows that the fungal community structure differed significantly in different regions (Anosim test, R = 0.6, *p* ≤ 0.001).

### 3.4. Co-Occurrence Networks and Assembly Mechanisms of Soil Microeukaryote Communities

The soil environment exhibits significant heterogeneity and harbors abundant and diverse biological communities [38]. As depicted in Figure 5, notable changes were observed in the co-occurrence networks due to differences in the sampling areas. The network’s topological properties revealed a high level of modularity in the symbiotic networks, consistently ranging from 0.598 to 0.835. This indicates that the networks formed distinct clusters or modules. The micro eukaryotic networks in the different study areas all formed distinct clusters or modules. Across various sampling areas, the proportion of positive correlations in co-occurrence networks significantly exceeded the proportion of negative correlations.

The differences among protist biological communities appeared to be less constrained by deterministic factors, as they occupied a narrower ecological niche (Figure 6A). Thus, fungal communities showed higher ecological niche indices than the protist communities. The narrower ecological niches than fungi indices of protists and fungi varied significantly across all seven sampling areas (Figure 6A). The niche index of the protist communities in DX was highly significantly lower than that of DYG (*p* ≤ 0.001), very significantly lower than those of LXL and TC (*p* ≤ 0.01), and significantly lower than that of LX (*p* ≤ 0.05). There were also significant differences in the niches of protist communities in LX compared with TC (*p* ≤ 0.05). Similarly, there were highly significant differences between the niche indices of the fungal communities of DYG and LX, LX and TC, and LX and WD (*p* ≤ 0.01). However, significant differences were found between DX and WD, and HN and LX (*p* ≤ 0.05).

Both protist and fungal communities in the entire region were assessed using the neutral community model (NCM) (Figure 6B). The relative contribution of random processes explained 90.0% and 67.9% of the variation in fungal and protist communities, respectively, indicating a lower influence of stochastic processes on fungal communities compared with protist communities. Additionally, the highest Nm (migration rate) value was found for protists (Nm = 434) (Figure 6B).

## 4. Discussion

The agroecosystem in the gap between the Tibetan Plateau and the Loess Plateau is strongly affected by climate. This has led to severe challenges to the sustainable development of agriculture and soil health protection in the region. Studies on agricultural soil organisms in the region provide new evidence for the monitoring of local soil quality and the improvement of the environment, as well as a reliable source of data to improve information on biodiversity between the Tibetan Plateau and the Loess Plateau. Soil communities play a crucial role in soil ecological processes, including decomposing organic matter, facilitating nutrient cycling, regulating microbial populations and community structure, and enhancing soil structure [39,40,41,42]. Protists and fungi, especially, contribute to ecosystem improvement and restoration in environmentally fragile areas [43]. Understanding the current status of soil organisms in agroecosystems can provide valuable insights for scientific management and sustainable development of croplands.

Specific soil ecological processes, land use changes, and agricultural production measures can strongly affect the spatial heterogeneity distribution of the soil microbial composition. Our high-throughput amplicon analysis revealed apparent changes in the composition and relative abundance of protists and fungi across different sampling environments. In particular, Alveolata, Rhizaria, and Amoebozoa showed the same pattern of relative abundance. Their wide range of lifestyles is the key to their evolutionary success in different ecological environments [44]. Ascomycota and Mucoromycota, which are ecologically important to soils and plants, demonstrated a high abundance, supporting their significance in local soils [45]. Gaining insights into the distribution and abundance of these groups, which play a vital ecological role in soil and plant ecosystems, can provide valuable information for making informed decisions concerning resource management, crop selection, and conservation efforts.

The analyses of the Shannon and Simpson indices revealed significant differences in the diversity of protists and fungi among sampling areas. Agricultural ecosystems have long been subject to periodic disturbances from human activities, including differences in crops, irrigation, and farming systems. Such disturbances are likely to have led to changes in soil microbial community diversity by causing changes in the soil environment. This suggests that the composition and distribution of protist and fungal communities are significantly influenced by the soil microenvironment [46,47,48]. Interestingly, it is possible that this significant difference is due to different weighting of species with the same abundance. The Shannon index emphasizes relative abundance, while the Simpson index emphasizes species richness [49,50,51]. The observed differences suggest that the soil microenvironment significantly influences the composition and distribution of protist and fungal communities. These variations should be considered when formulating agricultural practices, by taking into account the local soil conditions and environmental factors to promote beneficial soil microorganisms.

The beta analysis revealed that differences in geographic features and environmental conditions between the seven sampling areas play a crucial role in the formation of soil protist and fungal community structure. However, it is important to acknowledge the potential under- or over-representation of certain groups due to PCR amplification bias. Notably, the composition and structure of biological communities in the HN region differed significantly from other regions, aligning with expectations given the region’s perennially dry and rainy nature and its ecological fragility [52]. Soil environmental factors, such as pH, soil porosity, water content, and salinity, can all influence the composition and distribution of soil communities [53]. Protists depend on the layer of water that connects soil pores for movement, foraging, and reproduction, so their populations fluctuate in response to changes in pore water volume. This dependency on water significantly impacts the population size distribution of soil protists and contributes to variations in beta diversity [54]. Agricultural ecosystems shape the distribution of soil microorganisms within limited timeframes for adaptation, suggesting that the influence of farm crops on soil communities is not absolute. The results offer insights into the ecological characteristics of soil organisms under different environmental conditions, and such information can be leveraged to tailor farming practices to specific regions, thereby optimizing crop yield and overall ecosystem health.

Soil organisms form intricate ecological networks that can have positive, negative, or no significant impact on the species involved [55]. Our results revealed a strong relationship between environmental heterogeneity and species interactions. The predominance of positive correlations suggests reduced interspecific competition among species [56]. Moreover, the composition and structure of soil communities exhibit noticeable environmental heterogeneity, with farmland soil organisms being particularly susceptible to factors such as farming systems [57,58]. The complexity of interactions within soil organism communities is characterized by co-occurrence networks, which play a crucial role in maintaining environmental integrity and promoting plant well-being [59]. Unfortunately, we only studied the complexity of soil microbial interactions at a single time-point and could not explain the reasons for this result. However, a large number of studies have shown that different crop farming systems, plant growth status, and soil pollution all affect the complexity of microbial interactions in soil [60].

To investigate the assembly mechanisms of soil organism communities, we examined the interplay between stochastic and deterministic processes. Studies have indicated the importance of random processes in shaping fungal communities, with the neutral community model (NCM) demonstrating a better fit with higher diffusivity [35]. However, it is noteworthy that NCM may not fully explain the community assembly process, and other non-neutral effects are likely to be at play. Deterministic processes also exert significant influence on protist community assembly, particularly in communities with a narrow niche breadth [61]. Fungal communities exhibit higher ecological niche indices than protist communities, suggesting that the former are able to utilize more environmental resources, potentially leading to fungal dominance within the biome. Further research is needed to explore the extent to which niche theory and neutral theory can explain the mechanisms of species community structure formation and maintenance.

Understanding the complex interactions within soil organism communities, including positive and negative correlations, can aid in designing sustainable crop management strategies. These strategies can leverage the positive interactions among species to improve soil health, increase crop yield, and reduce the need for chemical inputs. Furthermore, investigating the assembly mechanisms of soil organism communities, considering both stochastic and deterministic processes, can shed light on effective approaches for managing soil biodiversity and fostering a more resilient agricultural system. The diversity of soil protozoa and fungi in farmland is of great significance in this context, as they contribute to soil quality improvement, nutrient supply, biological control of pests and diseases, enhanced plant growth, and pollutant degradation. Monitoring the soil biota can also serve as an essential tool for assessing soil health and guiding interventions to promote sustainable agricultural development.

Modern intensive agriculture has made great contributions to meeting food demand, but the contradiction between the environmental cost of agricultural intensification and the food security of sustainable development has become increasingly prominent. In order to solve this contradiction, an efficient and sustainable crop production method is urgently needed to increase the stability of the farmland ecosystem, thereby achieving increased crop yield and the efficient use of agricultural resources. The diversity of soil protists and fungi in farmland is of great significance to the sustainable development of agriculture and ecosystem protection. They can improve soil quality and provide a supply of nutrients supply; reduce the demand for pesticides through the biological control of pests and diseases; promote plant growth, thereby increasing farmland yield and quality; protect the environment, degrade organic pollutants, and stabilize harmful substances; and maintain biodiversity and maintain ecosystem health [62]. Monitoring the content of the soil biota can be used to assess soil health, which can help to target human interventions to promote sound agricultural development. The stability and development of farmland soil biodiversity has an important contribution to the sustainable development of agriculture and ecosystems. Therefore, knowledge of soil biodiversity is critical for maintaining ecosystem function in terms of carbon sequestration and emission reduction in farmland, soil fertility enhancement, and soil pollution control.

## 5. Conclusions

Understanding the diversity characteristics of soil protists and fungal communities will help us understand the role of microeukaryotes in the agricultural environment. The present study reveals a significant diversity of soil protists and fungal communities in different farmland habitats. NMDS and co-occurrence analyses provide evidence and insights into this diversity. In addition, our results highlight the importance of stochastic processes in shaping fungal communities and the important influence of deterministic processes on protist communities. These results are helpful to further explore the role of soil protists and fungi in soil ecosystem functions such as carbon sequestration and emission reduction. The stability and development of farmland soil biodiversity plays a crucial role in supporting sustainable agriculture and ecosystem health. Our research provides valuable data to understand the ecosystem functions of soil biodiversity, aiding in efforts related to carbon sequestration, soil fertility improvement, soil pollution control, and other critical aspects of sustainable agricultural practices. By integrating these findings into decision-making processes, stakeholders can work towards a more sustainable and ecologically balanced agricultural future.

## Figures and Tables

**Figure 1 microorganisms-11-02510-f001:**
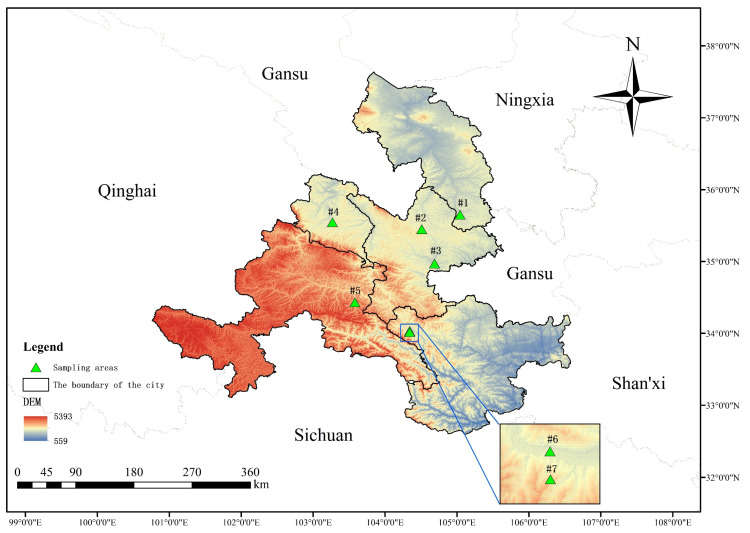
Sampling area map: #1 = Longxi samples, #2 = Dingxi samples, #3 = Huining samples, #4 = Linxia samples, #5 = Tanchang samples, #6 = Wudu samples, and #7 = Dayugou samples.

**Figure 2 microorganisms-11-02510-f002:**
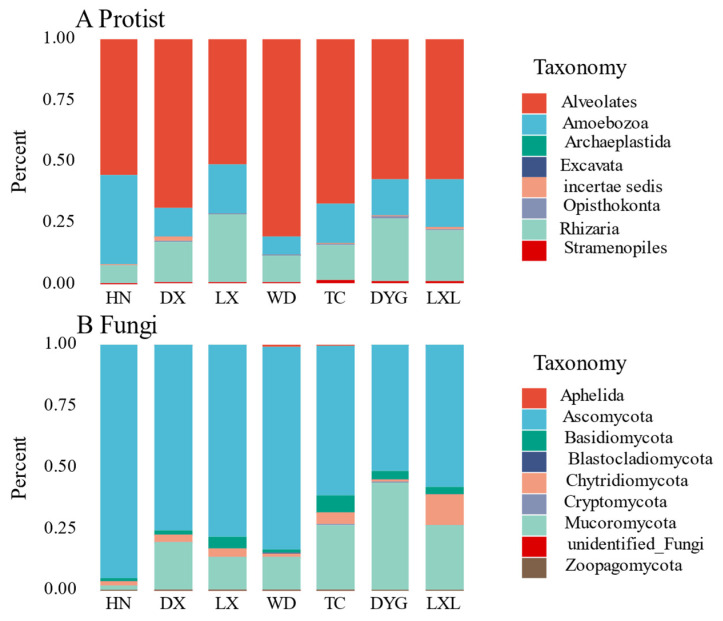
Mean of the relative abundances of the protists (**A**) and fungal (**B**) community composition for each sampling area. The size of the rectangular box represents the relative abundance. LX = Longxi samples, DX = Dingxi samples, HN = Huining samples, LXL = Linxia samples, TC = Tanchang samples, WD = Wudu samples, DYG = Dayugou samples.

**Figure 3 microorganisms-11-02510-f003:**
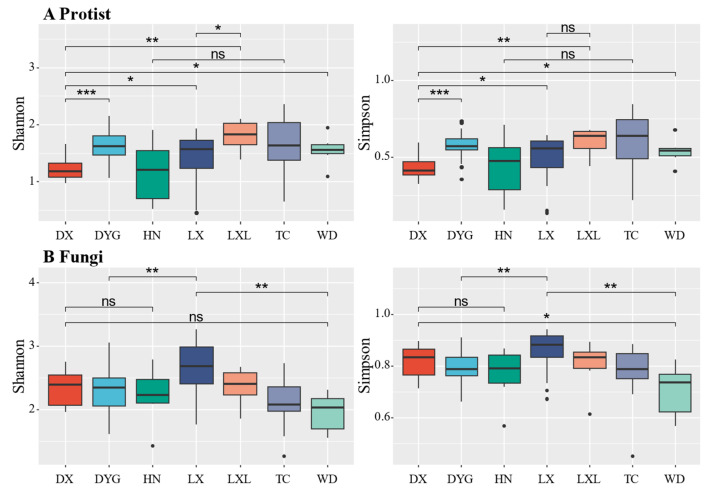
The alpha diversity of protists and fungi. “*” represents a significant difference at the 0.05 level, “**” represents a highly significant difference at the 0.01 level, and “***” represents an extremely significant difference at the 0.001 level. “ns” indicates no significant difference. LX = Longxi samples, DX = Dingxi samples, HN = Huining samples, LXL = Linxia samples, TC = Tanchang samples, WD = Wudu samples, DYG = Dayugou samples.

**Figure 4 microorganisms-11-02510-f004:**
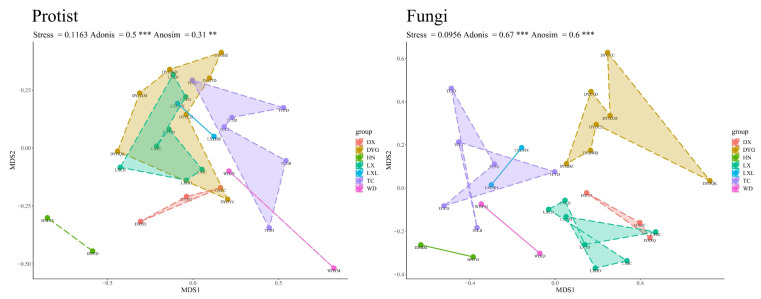
The beta diversity of protists and fungi. “**” represents a highly significant difference at the 0.01 level, and “***” represents an extremely significant difference at the 0.001 level. LX = Longxi samples, DX = Dingxi samples, HN = Huining samples, LXL = Linxia samples, TC = Tanchang samples, WD = Wudu samples, DYG = Dayugou samples.

**Figure 5 microorganisms-11-02510-f005:**
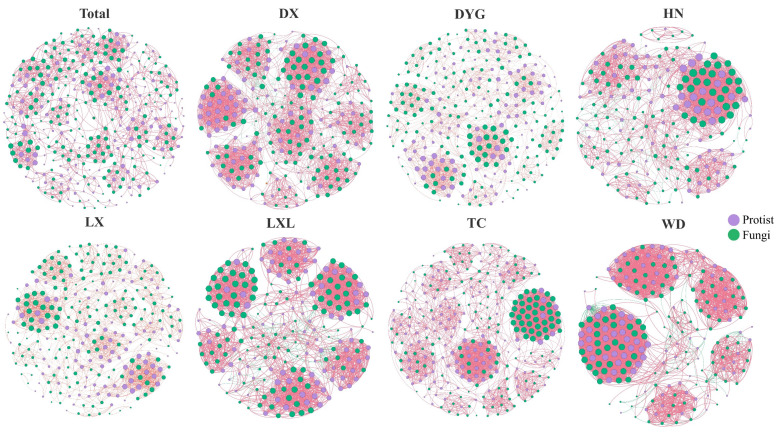
Interactions and changes between soil microbial groups in different sampling areas. The individual images show interactions between soil microbial groups and the images collectively show differences among the sampling areas. The red line indicates a positive correlation and the green line indicates a negative correlation. Purple indicates protists and green indicates fungi. LX = Longxi samples, DX = Dingxi samples, HN = Huining samples, LXL = Linxia samples, TC = Tanchang samples, WD = Wudu samples, DYG = Dayugou samples.

**Figure 6 microorganisms-11-02510-f006:**
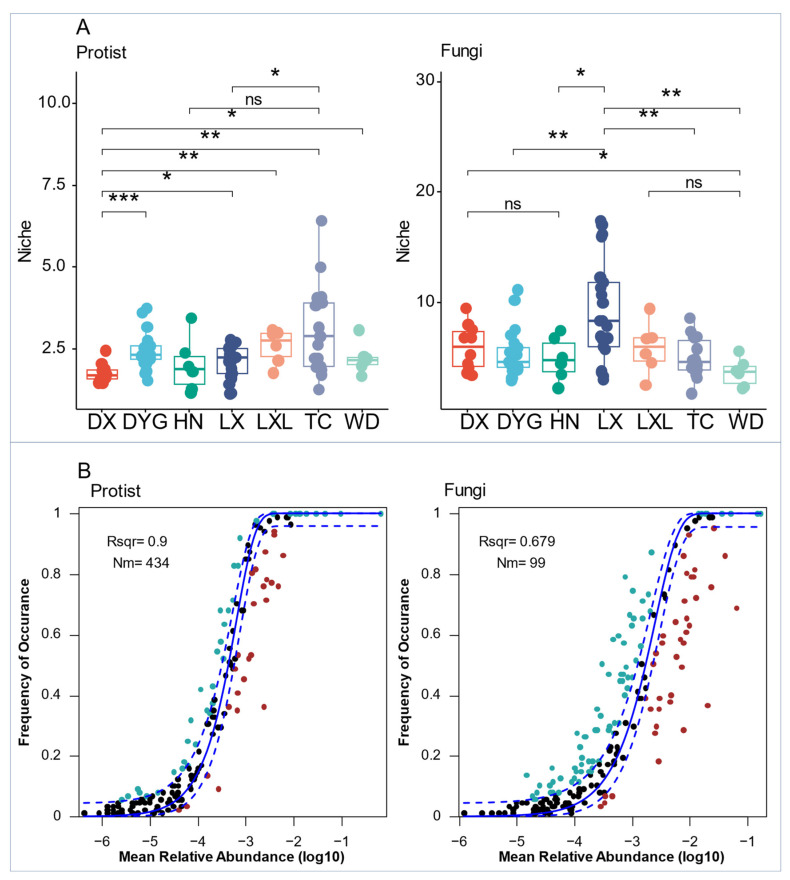
Ecological processes shape the protist and fungal communities in agricultural soils. (**A**) Comparison of mean niche breadth for protist and fungal communities among seven sampling areas. (**B**) The solid blue line is the best fit for the neutral community model (NCM), and the dashed blue line indicates 95% confidence intervals around the prediction. Blue and red spots mean species occurred more frequently or less frequently, respectively, than predicted by NCM. R^2^ (Rsqr) represents the fit to NCM. “*” represents a significant difference at the 0.05 level, “**” represents a high significant difference at the 0.001 level, and “***” represents an extremely high significant difference at the 0.001 level. “ns” indicates no significant difference. LX = Longxi samples, DX = Dingxi samples, HN = Huining samples, LXL = Linxia samples, TC = Tanchang samples, WD = Wudu samples, DYG = Dayugou samples.

**Table 1 microorganisms-11-02510-t001:** The ecological information of each sampling area.

Area Number	Sampling Area Short Name	Climate Type	Crop Type	Longitude (E)	Latitude (N)	Average Soil Temperature (°C)
#1	Longxi	Temperate monsoon	*Bupleurum chinense*	104°41′20″	34°58′16″	29.67
			*Phaseolus vulgaris*	104°41′20″	34°58′16″	24.33
			*Solanum tuberosum*	104°41′20″	34°58′16″	27.33
			*Allium.fistulosum*	104°41′21″	34°58′16″	28.67
			*Zea mays*	104°41′21″	34°58′16″	28.33
			*Glycine max*	104°41′19″	34°58′16″	35.00
			*Allium tuberosum*	104°41′21″	34°58′15″	31.00
#2	Dingxi	Temperate continental	*Solanum tuberosum*	104°30′32″	35°27′04″	30.00
			*Brassica oleracea*	104°30′38″	35°27′01″	28.00
			*Libanotis seseloides*	104°30′47″	35°26′59″	29.00
#3	Huining	Temperate monsoon	*Zea mays*	105°02′35″	35°39′05″	34.33
			*Sesamum indicum*	105°02′35″	35°39′05″	31.33
#4	Linxia	Temperate semi-arid and semi-humid	*Medicago sativa*	103°16′01″	35°32′52″	23.33
			*Lilium brownii*	103°16′19″	35°32′46″	30.83
#5	Tanchang	Temperate continental	*Zea mays*	104°20′34″	34°02′02″	25.00
			*Phaseolus vulgaris*	104°20′33″	34°01′59″	25.00
			*Cucumis sativus*	104°20′32″	34°01′57″	23.33
			*Cucurbita moschata*	104°20′32″	34°01′59″	24.00
			*Vigna unguiculata*	104°20′32″	34°01′59″	23.67
			*Raphanus sativus*	104°20′31″	34°01′59″	24.33
			*Capsicum annuum*	104°20′31″	34°01′59″	25.00
			*Solanum tuberosum*	104°20′33″	34°02′01″	27.67
#6	Wudu	Subtropical semi-humid	*Vigna unguiculata*	105°04′46″	33°19′42″	30.33
			*Zea mays*	105°04′46″	33°19′41″	30.33
#7	Dayugou	Plateau continental	*Brassica napusL*	103°34′47″	34°25′52″	24.00
			*Hordeum vulgare*	103°34′48″	34°25′53″	25.33
			*Vicia faba*	103°34′49″	34°25′55″	22.33
			*Pisum sativum*	103°34′48″	34°25′50″	21.33
			*Angelica dahurica*	103°34′48″	34°25′55″	25.67
			*Solanum tuberosum*	103°34′48″	34°25′56″	25.00
			*Triticum aestivum*	103°34′4″	34°25′58″	26.67

## Data Availability

Raw readings of all soil samples in this study have been deposited in the SRA of the NCBI database under accession number PRJNA1006017 or can be obtained from the authors.

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
