# Peer review of "Exploring Microeukaryote Community Characteristics and Niche Differentiation in Arid Farmland Soil at the Northeastern Edge of the Tibetan Plateau"

_microorganisms, 2023, doi:10.3390/microorganisms11102510_

Round 1
Reviewer 1 Report
I found the manuscript interesting and describing the new data on microeukaryotes structure and interactions in arid soils. I think that the presented manuscript is suitable for being published in the Microorganisms journal. However, I have one Major several Minor comments that should be fixed before acceptance. Major: Due to the fact that microeukaryotes communities were researched, I think that it should be mentioned in the title of the manuscript. In common perception of "microbial community" prokaryotes are included, thats wrong for research performed. Minor: Line 48 - space before references was missed. lines 74-75 - climatic parameters (heat and light energy) should be provided numerically. Lines 88-89 - should be no spaces before and after brackets. Lines 101-102 - please replace "should be.." with "were". Lines 105-106 - space after "(SDS)" should be added. Lines 124, 125, 127 - missed space before reference. Section 2.5 - the same comment: revise spaces close to references; URl addresses should be formatted as references.Figure 2 - If it is possible, please insert graphs into the manuscript as a pdf file for multiscale view. In the present form it is difficult to read the data presented. Lines 161, 168- the same situation with spaces and references Figure 3 - description of abbreviation and asterisks used is necessary in figure caption Figure 4 - the same comment as for figure 2. Lines 222, 267, 268, 280 - the same situation with spaces and references Line 243 - something going wrong with the reference :) Figure 6 - please use pdf and add descriptions for abbreviations Line 369 - spaces and references
Author Response
Dear Reviewer:
Thank you for your review comments on our manuscript entitled "Exploration of soil microbial community structure and ecological niche differentiation in arid farmland on the northeastern edge of the Tibetan Plateau" (microorganisms-2490824). These comments are valuable and helpful for us to revise and improve the paper. We have carefully studied these comments and made revisions in the hope of receiving your approval. The revised parts are marked in red in the thesis. The main revisions in the thesis and the responses to your comments have been described in detail in word.
We have made every effort to improve the manuscript and have made some revisions to it. These changes will not affect the content and structure of the paper.
We sincerely appreciate the reviewer's enthusiastic work and hope that the corrections are approved. Once again, thank you very much for your feedback and suggestions.
Sincerely,
。

Reviewer 2 Report
Exploring microbial community structure and niche differentiation in arid farmland soil at the northeastern edge of the Tibetan Plateau
The authors have made some advances on the manuscript and the figures are very nice. However, there are some issues that still need to be addressed particularly with repeatability, a clear and separate results and discussion sections, and data availability.
Key words: All words in the title are key words and the authors should consider removing repeated words. Suggest replacing community structure with something else.
Introduction: In general the introduction is well organized.
Line 64: Suggest changing fungal with fungi
Material and Methods: This section needs to be standalone meaning that it’s ok to site the methods but the readers should not need to go to the citation in order to repeat the experiment. So please cite the method and then briefly explain how it was done. Also, this is NGS data which needs to be deposited into a public open access repository so that other scientists can repeat this process. The data availability statement does not sufficiently state why the authors cannot deposit this information as suggested by MPDI data deposit requirements.
Section 2.1: Lines 73 to 75: This sentence is very vague and provides no concrete explanation of the physical properties of the area. For example, what does “sunshine time is sufficient” mean. This sentence could be removed.
Line 77: suggest removing the word “the” as it is not needed.
Line 80: suggest removing the word "the” before study and replacing it with “our”
Lines 82-93 need to be concise as this information is already on a table. Just make a statement that the samples are from geographically distinct regions with different climate conditions (Table 1, Figure 1).
Figure 1: I do not understand why there is a bar chart on the map. Is this a result? If it’s a result it would be more appropriate in the results section.
Line 102: suggest removing “ should be” and replace with “were”
Section 2.2: Currently this section is not repeatable. Describe in short the direct lysis method. How was the DNA pooled and concentrated? A kit? Alcohol precipitation? What were the PCR volumes, make and model of the thermocycler, volume of the PCR, thermocycle conditions for amplification. This is all missing for repeatability. How much DNA was used as template?
Section 2.3: Also lacking repeatability. If the authors completed the steps then describe in brief. If they sent this to a company, just state that all these steps were completed by the company. Also, NovaSeq6000 platform: what is the amplicon size? 100 bp, 200 bp, 250 bp. Single or paired-end reads?
Statistical analysis: The authors have some methodological information in the results section ( lines 157-160). The authors need to have a clear methods section, a clear results section, and a clear discussion section.
Line 129: analysis should be capitalized for consistency with the other subheadings.
Line 130: The datasets were subsampled to normalize the number of reads per sample. How was it subsampled, and at what depth was the data subsampled? Without knowing the depth of subsampling, it is not repeatable. Please add this information.
Lines 144-147: This information would be better under the figure 1 caption as its not a statistical analysis.
Results: The results are not clear in that there are both methods and interpretation in this section. In addition, the entire results section should be written in the past tense.
Figure 2: Unfortunately, the two red colors look the same. Is there a better color scheme? Also, I might have missed it but how was relative abundance calculated? This is important for repeatability as there is more than one way to calculate relative abundance. Please add it to the methods section.
Lines 154-155: This style of sentence should be avoided as it is verbose.
Alternatively starting from Line 153:
The relative abundance of soil protists (Figure 2 A) and fungal (Figure 2B) communities showed significant variation among the seven sampled regions.
The relative abundance of soil protists and fungal communities (Figure 1 A, B respectively)showed significant variation among the seven sampled regions.
As mentioned,: Lines 157-160 are methods not results.
Line 162: significantly lower? What is the p-value?
Line 166: suggest changing “is” to “was”. The results are in the past tense. (also see line 173-two places).
Lines 168-170: “Therefore, it is reasonable” is an interpretation. Just make factual statements and avoid interpretive wording.
Figure 3 caption needs more specificity. This is not all the soil community so state that it is for Protists and Fungi.
Line 179: suggest replacing “fungal” with “fungi”. Also, this is methods. Alpha diversity was inferred based on genus level taxonomy. All your analyses need to state at what level it was completed and should be in the method section.
Lines 191-200: Most of this section comes across as interpretation based on “it is worth noting that”. Please make result based statements.
Line 201: 3.3 needs to go on its own line.
Figure 4 caption is not specific enough. Also, suggest changing fungal to fungi in this case.
Line 206: “pronounced effect” is too vague. Use precise words in the results.
Line 209: What is the R value for the Anosim? This is very important to show for this test. Also, in general, the beta-diversity results are not clear. Just state the results directly. For example, lines 212-215 there are words like “it is obvious” and “This suggests that” which are interpretation. Also, the R value in this section is not clear which test this goes too?
Lines 221-225: Once again, most of this is methods and needs to be in the methods section.
Lines 230-232: Indicates is not a result but an interpretation.
Lines 234-237: Suggesting prevalent is not a result but an interpretation. Also, unless the authors measured the physical and chemical properties of the sampling area, I am not sure how they can make this statement. If the authors did measure these properties, then it should be added to the manuscript.
Line 239: Fungal should be changed to Fungi….and 240 suggesting is an interpretation.
Line 250: fungal should be changed to fungi.
Discussion section:
Lines 275-277: The authors state a common trend but what is the trend and why is that important.
Line 279-281: The authors support their expectation by stating that Ascomycota is dominate in plant tissue, but they tested soil. There are plenty of soil analysis showing this so a more appropriate support would be a soil paper reference.
Line 373-375: I don’t believe the data supports this statement as written. I think that in order to do all those things, we must understand the biodiversity within the soil so perhaps this statement just needs revised some.
Data Availability statement: This statement does not conform to MDPI and the authors must state why they cannot make this data publicly available in data repository. Better yet, the data should be deposited prior to publication.
The manuscript has some grammatical errors. In general, please review the methods (should be past tense), results should be more straight forward and should be past tense).
Author Response
Dear Reviewer:
Thank you for your comments concerning our manuscript entitled “Exploring microbial community structure and niche differentiation in arid farmland soil at the northeastern edge of the Tibetan Plateau” (microorganisms-2490824). Those comments are all valuable and very helpful for revising and improving our paper, as well as the important guiding significance to our researches. We have studied comments carefully and have made correction which we hope meet with approval. Revised portion are marked in red in the paper. Detailed responses are in Word.
Once again, we would like to thank you for your valuable comments and constructive criticism. Your comments will undoubtedly help us to improve our work. If you have any other suggestions or questions, please feel free to let us know.
Best regards,
[Haifeng Han]

Round 2
Reviewer 2 Report
Exploring microeukaryotes community characteristic and niche differentiation in arid farmland soil at the northeastern edge of the Tibetan Plateau
The authors have made some progress and have deposited their data. There are still some minor corrections throughout that will need to be adjusted prior to publication. Currently, the methods still need additional information to be repeatable, grammar issues need to be corrected, and figures should be stand-a-lone. The short IDs appear as abbreviations and should be defined in all figures if they are abbreviations. The authors should review the entire manuscript for similar issues pointed out below.
1) Please consider changing lines 73-95 to the following: Please review
2.1. Sampling Areas and Site Descriptions
Gansu Province is in the northwest of China at the intersection of three major plateaus (Loess Plateau, Qinghai Tibet Plateau and Inner Mongolia Plateau) and three major natural areas (northwest arid region, Qinghai-Tibet Plateau and eastern monsoon region). Its geological landforms and weather patterns are complex and diverse, and various geological forms such as plateau, mountain, desert and Gobi are intertwined. Previous studies have focused on the central region of the Tibetan Plateau, but our study area is located in the northeastern margin of the Tibetan Plateau, which complements the systematic study of the Tibetan Plateau. A comprehensive dataset comprising 87 soil samples was collected (Table 1), representing 29 different crop soils. These samples were geographically divided into seven sampling areas, as depicted in Figure 1.
2) Table 1: Based on the above information, I wonder why table 1 does not include the different crop soil types sampled in each area. If the authors mention this, it would be valuable to report as it is part of the ecological information. This would be of interest to the readers so please add.
3) Figure 1. As mentioned in previous review. The authors need to make each table and figure standalone so that the reader can easily understand all components of the figures. Please add information to Figure 1 legend. For example, consider something like the following:
Figure 1. Map indicating sampling locations in the northeastern margin of the Tibetan Plateau. The relative height of the bar chart represents the relative size of soil microbial community abundance. #1= LX samples, #2 = DX samples, #3 = HN samples, #4 = HN samples, #5 = TC samples, #6 = WD samples, #7 = DYG samples. Figure 1 was constructed using ArcView Gis 10.8 with the base bitmap downloaded from the National Catalogue Service of Geographic Information [21] and Geospatial Data Cloud [22].
4) The current sub-header (2.1) is outside the scope of this information. I would suggest a new subheading or modify the 2.1 sub-heading. Also, the new information added is best placed under Figure 1 so it is a stand-a-lone figure. Below is some suggested edits for this section (lines 100-109).
2.2. Soil sample collection and storage
Soil samples were collected in different research areas in August 2018 (Figure 1). Three 5 m × 5 100 m parallel quadrats were set up for each crop type. The surface soil (0 cm-5 cm) was collected strictly according to the five-point method of plum blossom for each parallel quadrat, and the fresh soil samples were stored in plastic fresh-keeping bags. Soil samples were stored at -20 °C to prevent DNA degradation. Soil temperature and geographic location was recorded at each collection point (Table 1).
5) 2.2. DNA Extraction and PCR Amplifications.
In this section, the author’s state that they processed the soil samples (30 samples) and pooled them together. For repeatability, the authors need to state how much soil sample was processed for each. What I mean is was the pooled DNA from each sample based on 2 grams of total soil extraction or some other amount? It is not possible to repeat this experiment without knowing how much soil sample was extracted per sample.
6) 2.3. High-Throughput Sequencing: This section needs all the information to be repeatable. This is still the main use with this manuscript. I suggest something like this, but the author still did not state if the sequencing was 250 bp pair-end reads V2 chemistry to something different? Was this a single read run? What was the length of the reads? What was the chemistry of sequencing (v2 or V3).
In short, PCR products were purified and visualized on a 2% agarose gel. Sequencing libraries were prepared from purified PCR fragments using a TruSeq® DNA PCR-Free Sample Preparation Kit (Illumina, USA) and quantified using a Qubit@ 2.0 Fluorometer (Thermo Scientific, USA). Prepared DNA libraries were pooled and sequenced on an Ion NovaSeq6000 platform with XXX bp vX chemistry. All these steps were completed by the Novogene Science and Technology Company (Beijing, China).
7) Lines 148-149: Please consider the following:
Both diversity parameters, species diversity (Shannon-Weaver H′) and dominance index (Simpson C′), of protists and fungal communities were calculated using R software with the "vegan" package, and the significance was tested using Wilcoxon test [31].
8) Lines 152-153: Consider the following:
The alpha diversity of protists and of fungi in the samples was assessed as the genus level.
9) The figure legend still needs some additional explanation. As a reader, I do not know what HN, DX, etc. are other than soil samples. Looking at Table 1, the authors state these are short ID, but I have not found within the manuscript was the full ID is for these. It would be more transparent to have the full ID in the figure header legends. For example: HN = XXXX, DW = XXX.
Figure 2. Protists and fungal community composition according to sampling areas. The size of the rectangular box represents the relative abundance. HN = XXXX, DW = XXXX, ect….
10) Line 189-: Sentence needs to be in the past tense and if this information has not been published, there should be no citation. Consider the following:
As for fungal communities, there were significant differences in the fungal communities in the seven regions (Figure 2 B). Asomycota was the dominant group in all seven regions, accounting for more than 50% of the entire fungal community, with the highest relative abundance of species in the HN region, accounting for more than 90%. Ascomycota was clearly the dominant fungal group in all seven regions, comprising over 50% of the entire fungal community. Moreover, it exhibited the highest relative abundance in the HN region, accounting for more than 90% of total fungal species. Mucoromycota was also detected in all regions, albeit with a significantly lower proportion than Asomycota, except in DYG where the two groups exhibited nearly identical relative abundances. Aphelida species were only found in WD and TC but was rare in abundance. The relative abundances of Aphelida, Cryptomycota, unidentified_Fungi and Zoopagomycota were very rare in all sampled regions.
Other quick comments:
Line 224 (obviously), Line 264 (it is easy to conclude) are too vague for result, just state the results without these kinds of phrases.
All figures: The short IDs are abbreviations and should be defined in the figure legend for greater clarity and for having the figures stand-a-long. Please review all.
There are some additional grammatical errors in the results and within the discussion. Please consider a careful review of these sections.
The authors should take their time on correcting the manuscript.
Author Response
Dear reviewers,
Thank you for once again suggesting revisions to our manuscript entitled "Exploring the structure and ecological niche differentiation of soil microbial communities in arid farmland on the northeastern edge of the Tibetan Plateau" (microorganisms-2490824). These comments were very helpful to us in revising and improving the paper, and were also important guidance for our research. We have carefully studied these comments and made revisions in the hope of getting your approval. The revised parts are marked in red in the thesis. The main revisions of the paper are in word.

Round 3
Reviewer 2 Report
The authors have made great improvements to this version of the manuscript. It is better organized, more concise, the data has been deposited, and the figure legends are more informative.
I only have a few minor corrections.
Line 11: I believe experiences should be changed to experienced. Please consider.
Line 136: the authors should add the word of and remove the word a.
Consider: The mixture of PCR products was purified with....
Line 177: Consider adding a space between the words patterns and were. Currently it is patternswere
Line 204-205: This sentence is still awkward. Please review.
Line 317-319: This sentence is still awkward. Please review.
There are one a few sentences that need to be revised.
Author Response
Dear reviewer,
Thank you for once again reviewing our research paper and providing valuable comments and suggestions. We appreciate your recognition of our research and would like to respond to the questions you have raised.
Line 11: I believe experiences should be changed to experienced. Please consider.
Thank you for your comments on our writing tenses. We have changed the word "experiences" to "experienced".
Line 136: the authors should add the word of and remove the word a.
Consider: The mixture of PCR products was purified with....
Thank you for your suggestions about our writing. We have carefully considered your suggestion and revised it to "The mixture PCR products was purified with GeneJETTM Gel Extraction Kit (Thermo Scientific).
Line 177: Consider adding a space between the words patterns and were. Currently it is patternswere
Thank you for your suggestion. We have re-added the spaces as you suggested and have now changed them to patterns were ..... .
Line 204-205: This sentence is still awkward. Please review.
Line 317-319: This sentence is still awkward. Please review.
Thank you for your writing suggestions. lines 204-205 we have changed to " Alveolates were a significant proportion in all regions, accounting for 50% in each region, with the highest abundance being observed in the WD and the lowest in the LX. " Lines 317-319 have been deleted, and we assure you that this will have no impact on the integrity of the article's content.